# The neutral rate of whole-genome duplication varies among yeast species and their hybrids

S. Marsit [1,2,3,4 ✉], M. Hénault[1,2,3,4], G. Charron [1,2,3], A. Fijarczyk [1,2,3,4] & C. R. Landry [1,2,3,4 ✉]

Hybridization and polyploidization are powerful mechanisms of speciation. Hybrid speciation often coincides with whole-genome duplication (WGD) in eukaryotes. This suggests that WGD may allow hybrids to thrive by increasing fitness, restoring fertility and/or increasing access to adaptive mutations. Alternatively, it has been suggested that hybridization itself may trigger WGD. Testing these models requires quantifying the rate of WGD in hybrids without the confounding effect of natural selection. Here we show, by measuring the spontaneous rate of WGD of more than 1300 yeast crosses evolved under relaxed selection, that some genotypes or combinations of genotypes are more prone to WGD, including some hybrids between closely related species. We also find that higher WGD rate correlates with higher genomic instability and that WGD increases fertility and genetic variability. These results provide evidence that hybridization itself can promote WGD, which in turn facilitates the evolution of hybrids.

---

[1] Institut de Biologie Intégrative et des Systèmes, Université Laval, Québec, QC, Canada. [2] Regroupement Québécois de Recherche sur la Fonction, l'Ingénierie et les Applications des Protéines, (PROTEO), Université Laval, Québec, QC, Canada. [3] Département de Biologie, Université Laval, Québec, QC, Canada. [4] Département de biochimie, microbiologie et bio-informatique, Université Laval, Québec, QC, Canada. ✉email: souhir.marsit@gmail.com; christian.landry@bio.ulaval.ca

Whole-genome duplication (WGD) is an important evolutionary force in eukaryotes[1–6]. Nearly one-third of contemporary vascular plant species have undergone or have ancestors that have undergone WGD[7]. Among animals, polyploidy is less frequent but well-known cases occurred in parthenogenetic and hermaphroditic animals[8]. In vertebrates, polyploidy resulting from WGD is much less frequent[1]. However, two rounds of WGD occurred in the ancestor of the vertebrate lineage leading to humans[9,10]. Important events in the evolution of fungal diversity have also been associated with genome doubling[2]. A WGD occurred about 100–150 million years ago during the evolution of *Saccharomyces* yeasts[11–14].

Autopolyploid and allopolyploid yeasts have been described in anthropic and some natural environments[15–24]. In the budding yeast *Sacchromyces cerevisiae*, most of the natural isolates are diploid. However, polyploid isolates (3–5*n*) (11.5% of 1011 yeast genomes) are enriched among domesticated strains, suggesting that some human-related environments have had an effect on triggering changes to higher ploidy levels or favoring the propagation of polyploid strains[21]. Bakery, Ale beer and clinical strains of *S. cerevisiae* are frequently auto-tetraploid[16,18–20]. Polyploidy is also commonly observed in recent human associated[13,22–25] and synthetic[26–28] yeast hybrids, suggesting that allopolyploidization might not be a rare mechanism of yeast hybrid speciation[13]. Many industrial strains are allopolyploids resulting from hybridization between different *Saccharomyces* species[13,22–25,29]. The lager beer yeast *S. pastorianus* was shown to be an allopolyploid hybrid between an ancestor of the ale beer yeast *S. cerevisiae* and *S. eubayanus*[30]. Allopolyploid wine and cider yeasts have also been reported, including *S. cerevisiae × S. kudriavzevii*, *S. cerevisiae × S. uvarum* and *S. cerevisiae × S. kudriavzevii × S. uvarum* hybrids[31–36]. Compared with the parental strains, these hybrids have several advantages for wine and beer fermentation, including increased tolerance to various stresses and increased fermentative performance[22–24,26,29,37]. In fact, only a few cases of nonsynthetic *Saccharomyces* homoploid hybrids (same ploidy as parents) have been reported[38,39].

While significant progress has been made in the last decade in understanding the consequences of WGD[1,40], the factors that favor its evolution are less well studied[41]. A large body of work, mostly on plants, showed that hybrid speciation often coincides with WGD[1]. A long-standing hypothesis suggests that parental genetic divergence influences the probability of WGD in hybrids[42]. Combining two diverged genomes in the same cells would increase the rate of genomic changes and WGD, which would in turn enable hybrid maintenance on the long term[43]. Recent studies showed that homoploid hybrid plants tend to be derived from parents that are less evolutionarily divergent than parents of polyploid hybrids[44–49]. However, the role of genetic divergence as a driver for WGD has been debated[44–49]. The debate comes from the fact that the positive correlation between parental divergence and polyploidization could be related to the success of hybrid establishment rather than the rate of WGD itself. Allopolyploids could be more likely to be maintained because WGD in hybrids increases fitness by restoring fertility[42,50–52] and accelerating adaptation[53]. On the contrary, auto-polyploids frequently have reduced fertility due to multivalent pairing during meiosis[54]. These effects combined would increase the frequency of WGD in hybrid species. Testing these alternative models, increased rates of WGD versus enhanced adaptive potential of hybrids following WGD, requires removing natural selection from the equation.

In this work, using yeast as an experimental model, we show that WGD is more likely to occur in some parental lineages, but can also be triggered by hybridization through the combination of some genotypes. We also find that higher WGD rate correlates with higher genomic instability, and that WGD subsequently increases fertility and genetic variability. Together, these results provide evidence that hybridization itself can promote WGD.

## Results

**Hybrids and intra-lineage crosses WGD rate**. Using yeast as an experimental model, we investigated whether hybridization can accelerate WGD. We measured the rate of WGD in 1304 independent yeast lines from 15 crosses following a protocol for mutation accumulation (MA). MA lines were evolved through repeated strong bottlenecks to remove the confounding effect of natural selection. We used crosses over different levels of parental divergence from intra-lineage to interspecific crosses (Fig. 1a). Natural yeast isolates representing three incipient species of the wild yeast *Saccharomyces paradoxus* and its sister species, *S. cerevisiae*, were used. The *S. paradoxus* lineages (*SpA*, *SpB* and *SpC*) exhibit up to 4% nucleotide divergence, and 15% with *S. cerevisiae* while their genomes remain largely co-linear[55,56] (Fig. 1a). We previously generated 864 lines[52,57] and here generated 288 new lines (3 crosses) using the same procedure (Supplementary Table 1). All crosses were evolved for roughly 770 generations (Fig. 1a). We also examined 152 MA lines derived from a homozygous diploid *S. cerevisiae* strain propagated for ~2062 generations[58]. The MA lines were classified in terms of nucleotide divergence in four types: very low (VL:VL_B = *SpB × SpB*, VL_C = *SpC × SpC*, VL_A = *SpA × SpA* and VL_S = *Scer × Scer*); low (L = *SpB × SpC*); moderate (M = *SpB × SpA*) and high (H = *SpB × S. cerevisiae*) (Fig. 1a, Supplementary Table 1).

We measured the change in ploidy of all lines by quantifying DNA content using flow cytometry. We detected WGD (from diploid to tetraploid) in 7 crosses among all 4 types (Fig. 1b, Supplementary Fig. 1). Thirty-two tetraploid lines were identified in total (Supplementary Figs. 1 and 2). Using whole-genome sequencing, we found that for all tetraploids, the frequency of parental alleles across the genomes is roughly 50%, confirming that both parental genomes have been entirely duplicated (Supplementary Fig. 3). This shows that within 770 generations, spontaneous WGD occurred in 0–11% of the populations among the different crosses (Fig. 1b, c, Supplementary Table 2).

WGD occurred in the three hybrid types (L, M, and H) and in only one of the intra-lineage cross types (VL_C). Since WGD is observed exclusively in the three VL_C crosses (VL_C1 (11%), VL_C2 (6%), VL_C3 (9%)) and not the other VL crosses, we set out to investigate if *SpC* strains are intrinsically more prone to WGD. To test this, we analyzed the ploidy of more than 300 wild North American *S. paradoxus* isolates belonging to the three *SpA*, *SpB* and *SpC* lineages[55]. Consistent with the observations in the MA lines, *SpC* is the only lineage from which a natural tetraploid strain was found (Supplementary Fig. 4). We have also previously shown that some *SpC* haploid strains are prone to spontaneous diploidization (1*n*–2*n*)[52]. As a consequence, almost half of the L1 and L2 lines are triploid at the initial timepoint (Supplementary Fig. 1). They most likely arose from mating between haploid *SpB* strains and diploidized competent for mating *SpC* strains that arose from haploid stocks[52]. WGD also occurred in L1 and L2 hybrids with lower frequency than in VL_C (3% and 2% respectively). These hybrids share the same *SpC* parents with VL_C crosses and the same *SpB* parents with VL_B crosses, suggesting that the WGD rate observed in L1 and L2 hybrids is potentially a trait inherited from the *SpC* parental subgenome.

WGD occurs in hybrids but not in all biological replicates; only one of the two replicate crosses of the (M) and (H) crosses show WGD (M1 (4%) and H2 (6%)), suggesting that certain genotypes or combinations of genotypes are more prone to WGD (Fig. 1b, c).

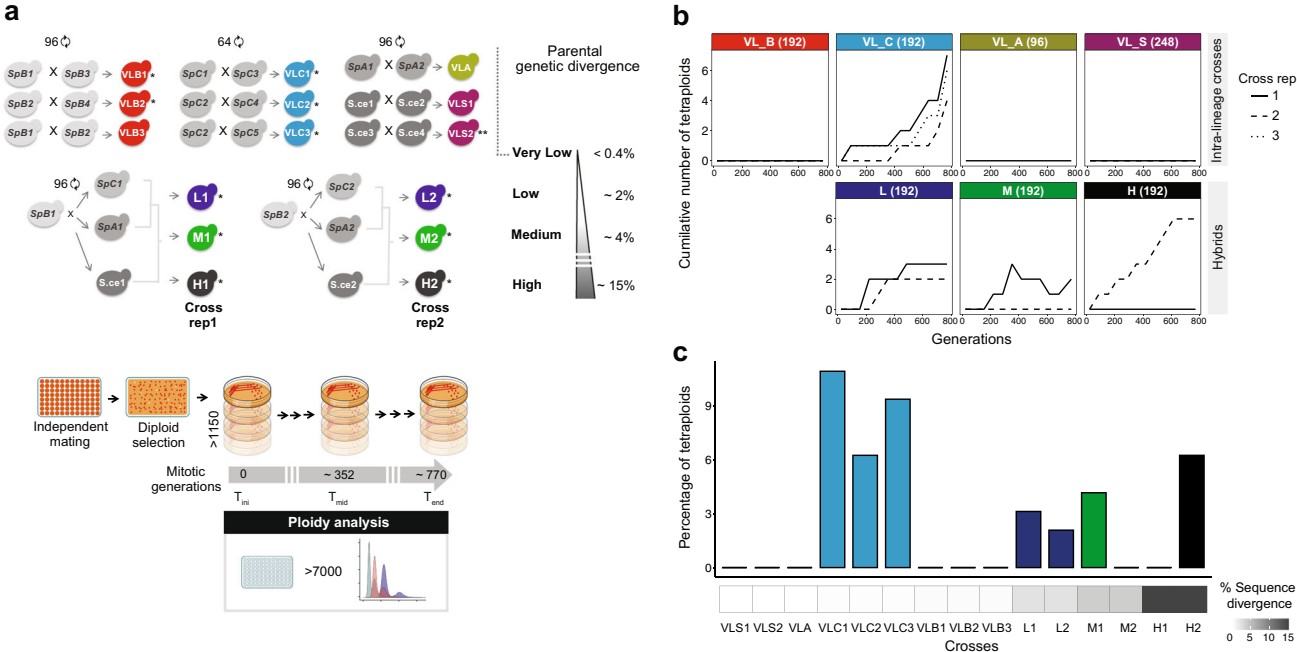

**Fig. 1 Whole-genome duplication is genotype and cross specific. a** Crosses among *S. paradoxus* incipient species (VL_B, VL_C, VL_A, L, and M), among *S. cerevisiae* isolates (VL_S) and between these two distant species (H). Most types of crosses involve two to three biological replicates, and each individual cross was performed independently 48–96 times to represent independent matings. We also included 152 MA lines derived from a homozygous diploid *S. cerevisiae* strain (VL_S2) propagated for ~2062 generations[58]. Crosses are labeled with a star according to their source *[52,57] and **[58]. The 1304 lines were evolved under relaxed selection. Mitotic propagation was performed through repeated single-cell bottlenecks and ploidy was measured using flow cytometry at different generation timepoints. **b** Cumulative number of tetraploid lines observed along the experiment. Numbers in parentheses represent the number of biologically independent studied lines. **c** Whole-genome duplication rate differs among crosses (VL_C1 ($n = 7$), VL_C2 ($n = 4$), VL_C3 ($n = 6$), L1 ($n = 3$), L2 ($n = 2$), M1 ($n = 4$), H2 ($n = 6$), $n$ represents the number of biologically independent tetraploid lines).

Because the parental strains used in M1 (*SpB1* and *SpA1*) and H2 (*SpB2* and *Scer2*) crosses are also involved in VL_B, VL_A, and VL_S intra-lineage crosses in which WGD did not occur, our results cannot be explained by the ploidy instability of these parental strains, and thus support the hypothesis that WGD rate can be increased by hybridization.

Genome doubling occurred at different timepoints after mating; some of them emerged early, before the 90th generation, while others emerged after more than 680 generations (Fig. 1b and Supplementary Fig. 2). Two of the 32 tetraploids went extinct before the end of the experiment (L1_31 and H2_38). The remaining 30 tetraploids evolved from a few generations to more than 680 generations. We find that tetraploidy can be slowly reverted. One of the identified H2 tetraploids (H2_43) shows progressive reduction in ploidy from $4n$ to about $2.8n$, 418 generations after WGD (Supplementary Fig. 2). Genome sequencing of this strain confirms several aneuploidies with copy number reductions for three large chromosomes (Chromosomes V, XII, and XIV) (Supplementary Figs. 5 and 6). However, the copy number reduction of these three large chromosomes cannot fully explain such a decrease in ploidy (from $4n$ to $2.8n$). These results could be explained by heterogeneity among isolated aneuploid colonies from glycerol stock, probably due to the high genomic instability of hybrids[52]. Two of the M1 tetraploids (M1_32 and M1_40) also show rapid ploidy change from tetraploid to diploid. However, mixed colonies containing $4n$ and $2n$ strains in M1 lines have been detected in a previous study and could explain this observation[52] (Fig. 1b, Supplementary Fig. 2).

**WGD impact on fertility**. Darlington[54] hypothesized that there should be an inverse relationship between the fertility of a diploid

hybrid and that of a tetraploid to which it gives rise (Fig. 2a). He reasoned that at low parental divergence, although a diploid cross would be fertile, the corresponding tetraploid would show low fertility because pairing could occur between any pair of the four homologous chromosomes, causing multivalent pairing and uneven segregation. At high parental divergence, the opposite would be expected. Diploid hybrids should be sterile due to the failure of homeologous chromosome pairing, but allopolyploids should be fertile due to consistent bivalent pairing between identical duplicated chromosomes at meiosis (Fig. 2a).

Previous studies from us and others confirmed that WGD can restore hybrid fertility[50–52]. However, how WGD affects the fertility of intra-specific crosses needs to be further tested. To test the first prediction of Darlington, we measured the fertility of 12 VL_C and all L ($n = 5$), M1 ($n = 4$) and H2 ($n = 6$) independent lines before and after WGD (the fertility of 8 of the L, M, and H tetraploids were previously described[52]). One typical proxy for fertility in yeast is the percentage of viable spores after meiosis[59]. As expected, most allotetraploid hybrids have a significantly increased spore viability compared to diploids that display low (L, M diploids) to almost null fertility (H diploids) (Fig. 2b, Supplementary Data 1). The only exception is observed for one tetraploid (H2_43) that lost its ability to sporulate. Contrary to what Darlington hypothesized and in agreement with previous studies in flowering plants[60], intra-lineage crosses (VL_C1 and VL_C2 lines) show no difference between their diploid and tetraploid state in almost all tested lines, with both ploidy levels showing very high fertility (Fig. 2b). On the contrary, VL_C3 lines show increased fertility after WGD, while VL_C lines that remain diploid show no change (Supplementary Fig. 7). These results suggest that there are genetic differences between the two parental *SpC* strains of the VL_C3 cross that decrease F1 fertility

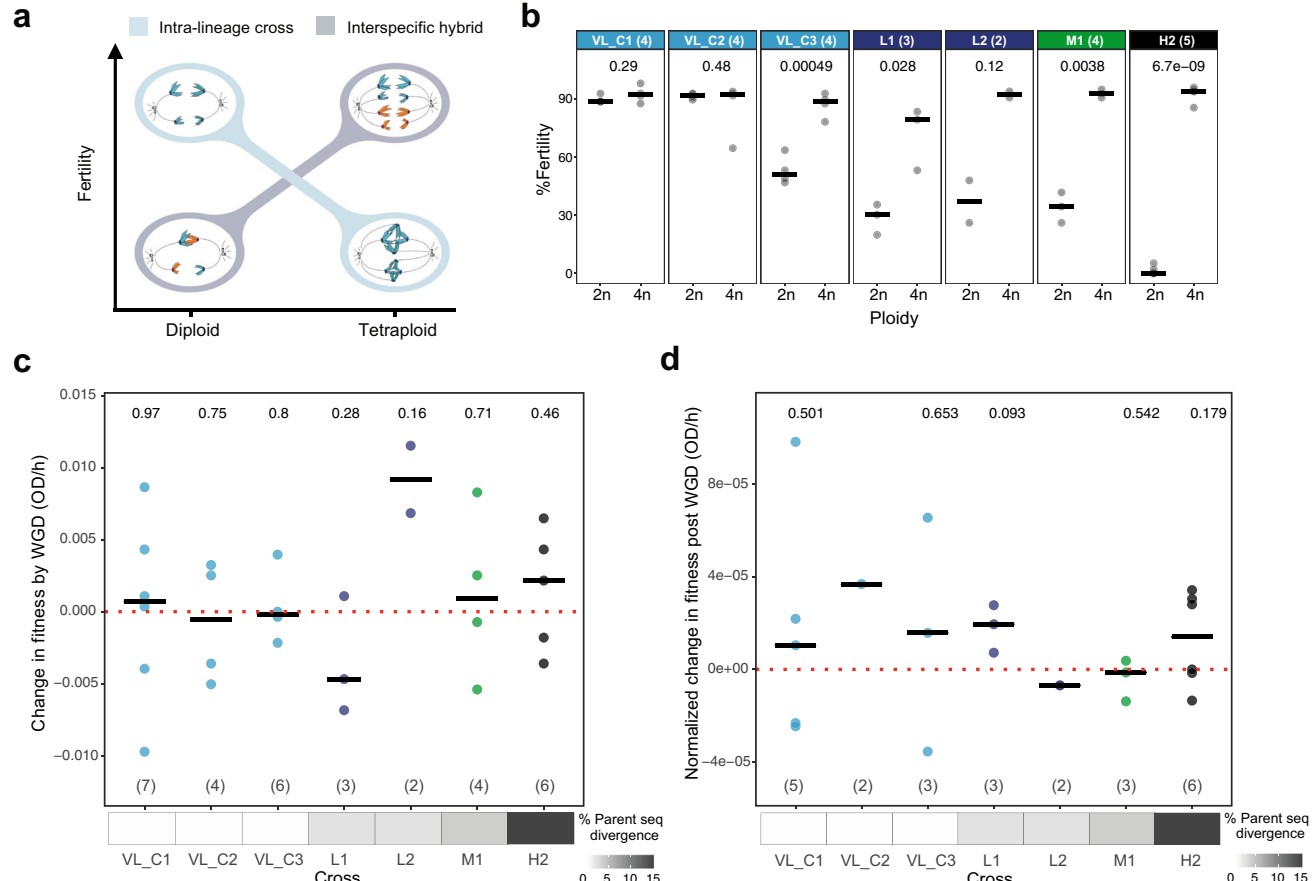

**Fig. 2 Whole-genome duplication increases fertility and has no systematic effect towards fitness gain or loss in intra-lineage crosses and hybrids.**
**a** Darlington hypothesis predicts an inverse relationship between the fertility of a diploid hybrid and that of a tetraploid to which it gives rise. **b** Impact of whole-genome duplication (WGD) on fertility. Fertility here is measured using as proxy the percentage of viable spores after meiosis. P values from a two-sided paired t-test are shown. **c** WGD has no systematic effect toward immediate fitness gain or loss when it happens. Change in fitness by WGD is calculated as the difference between the maximum growth rate before and after WGD. **d** Tetraploid lines show gains and losses of fitness at the end of the experiment. Change in fitness post-WGD is calculated as the difference between the maximum growth rate after WGD and at the end of the experiment divided by the number of evolved generations following WGD to the end of the experiment. Medians are shown by horizontal bars. P values from a two-sided t-test are shown to evaluate if the change in fitness is significantly different from zero in (**c**) and (**d**). Numbers in parentheses represent the number of biologically independent tetraploid lines.

and which is restored by WGD. Indeed, these two parental genomes are more divergent compared to VL_C1 and VL_C2 (+2% and +4% nucleotide differences, respectively).

**WGD impact on fitness.** Recent studies showed that polyploids are more successful than their diploid parents because they undergo significantly faster adaptation, presumably because of their access to more mutations that lead to increased phenotypic diversity[40]. However, this could also increase the rate of deleterious mutations. To first test how WGD affects fitness under relaxed selection, we measured the maximum growth rate of the 32 lines that underwent WGD at four timepoints: soon after mating, before and after WGD, and at the end of the experiment (Supplementary Fig. 8 and Supplementary Data 2). For comparison, we also measured the growth rate of 5 independent diploids from each cross after mating, in the middle and at the end of the experiment (Supplementary Fig. 8 and Supplementary Data 2).

Our results show that spontaneous WGD leads to both gains and losses of fitness, such that WGD has no systematic trend toward fitness gain or loss in these conditions (Fig. 2c) (paired t-tests, (Supplementary Fig. 9a)). Evolved tetraploids in the majority of crosses also show gains and losses of fitness at the end

of the experiment (Fig. 2d and Supplementary Fig. 9b). However, two of the H2 hybrids (H2_43 and H2_57) show respectively a markedly increased and decreased fitness (Supplementary Fig. 9b), suggesting that these hybrid lines had accumulated genomic changes that highly affected their growth rates and supporting the idea that allopolyploidy could increase phenotypic diversity.

**Higher WGD rate correlates with higher genomic instability.** It has been suggested that hybridization may increase the probability of WGD because combining two diverged genomes together would upset the course of cell division[43]. Newly formed hybrids indeed have an increased rate of alterations in the genome (genomic instability, GIN)[22–24,26,61,62] and WGD is one of the common GIN hallmarks in cancer cells[63]. However, little is known about the possible role of GIN in increasing the rate of WGD in hybrids. Our results show a higher rate of WGD in VL_C and L crosses, and only one of the two biological replicates of M (M1) and H (H2) hybrids. This result could be due to an overall increased GIN in crosses showing higher rates of WGD.

To test this hypothesis, we sequenced the genomes of (n = 33–39) randomly chosen diploid lines from VL_B (1 and 2), VL_C, M, and H crosses, as well as (n = 34) diploid and (n = 36) triploid lines from L crosses soon after mating and at the end of

the experiment. We measured the rates of aneuploidy (chromosome gain and loss) and loss of heterozygosity (LOH) resulting from mitotic recombination within chromosomes, two typical GIN hallmarks[64,65]. We looked at the diploid lines of these crosses, reasoning that crosses with more intense GIN would also have more unstable diploid lines.

Our results indicate that crosses showing higher rates of WGD manifest different hallmarks of increased GIN. For instance, diploid intra-lineage VL_C crosses showing the highest rates of WGD also exhibit the highest aneuploidy rate compared to VL_B crosses and to M and H hybrids (two-sided Mann–Whitney–Wilcoxon test, $P = 1.4e-10$, $P = 7.7e-06$ and $P = 1.1e-07$, respectively, Fig. 3a, b, Supplementary Fig. 10, 11). These crosses are trisomic or tetrasomic and have one or two additional copies of chromosome XII inherited from their parental strains (Supplementary Fig. 12). The same result is observed for L hybrids, which share the same $SpC$ parents with VL_C crosses (two-sided Mann–Whitney–Wilcoxon pairwise comparisons, $P = 0.0046$ and $P = 0.039$ respectively to VL_B crosses and H hybrids, Fig. 3a, b). This supports previous observations that the presence of extra chromosomes increases genomic instability[66] and could potentially trigger WGD by having additional chromosomes forming bridges that can prevent cytokinesis.

Our results also show that all hybrids exhibit increased GIN compared to the intra-lineage crosses VL_B and VL_S2[67] (Supplementary Table 2). Hybrids show significantly higher rates of whole chromosome loss (Fig. 3b, Supplementary Figs. 10, 11) and a lower LOH frequency (Fig. 3c). This is consistent with a lower efficiency of mitotic homologous recombination between homeologous chromosomes, which show a higher nucleotide divergence[68]. Aneuploidy frequencies of all 16 chromosomes indicate that chromosome XII is the most unstable chromosome among hybrids (Fig. 3b). This chromosome contains the rDNA locus, which is a large tandemly repeated sequence of 9 kb encoding for ribosomal RNA, one of the most unstable parts of the genome due to the repeated recombinations and copy number variation[69,70]. The high instability of this chromosome in hybrids could be a testimony of increased genomic instability.

Finally, we find that the replicates within crosses M and H that show higher rates of WGD (M1 and H2) manifest more hallmarks of GIN. The two replicates of interspecific hybrids, H1 and H2, show different LOH size ranges. The H2 hybrid lines (6% WGD) have a significantly longer tract of interstitial LOH segments than H1 lines (0% WGD) (two-sided Mann-Whitney-Wilcoxon test, $P = 0.0032$, Fig. 3d). The lengths of these regions argue against a classical gene conversion mechanism that would in general lead to smaller LOH segments. Instead, they suggest a higher frequency of LOH caused by aberrant chromosomal segregation events and/or break-induced replication events[24]. Long-range LOH events have been described to be more frequent under replication stress and to have a greater contribution to GIN in cancer cells[71,72]. Finally, M1 (4% WGD) and M2 (0% WGD) hybrids show a significant difference in rates of line loss. Much higher line decline has been observed in M1 hybrids, suggesting that these hybrids have higher GIN leading to the frequent segregation of highly deleterious variants compared to M2 hybrids[52].

**WGD increases genetic diversity**. Increased GIN as a consequence of polyploidy was previously observed[1,40] but not without the confounding effect of natural selection, which biases the set of observable mutations. We therefore investigated whether lines that went through WGD display higher genomic variability than diploids within the same cross. If these genomic

changes occurred after WGDs this would suggest that WGDs actually increase the GIN. We find that frequencies of aneuploidy in tetraploids are up to four times those of diploids (two-sided Mann–Whitney–Wilcoxon test, $P = 7.3e-05$ Fig. 4a, Supplementary Fig. 13). Different patterns of chromosome gain and loss are observed in tetraploids compared to diploids (Figs. 3b and 4b). Chromosome loss, consistent with previous studies[21,62,73,74] (for review[75]), more frequently affects the smallest chromosomes (I and III) in diploid hybrids, probably because it affects a smaller number of genes. However, larger chromosome losses (II, V, and VIII) are observed in tetraploids, suggesting a potentially less deleterious effect of their loss due to the multiple copies present in the cell. Furthermore, these chromosome losses occurred most likely after WGD. Indeed, if a chromosome loss occurs before WGD then aneuploid lines would be homozygous for that chromosome. Allele frequency analysis of tetraploid lines shows changes in allele frequency of aneuploid chromosomes but all of them remain heterozygous (Supplementary Fig. 6). These spontaneous aneuploidies have consequences on fitness. Two allopolyploids (H2_43 and H2_57) respectively show the most increased and decreased growth rates after WGD (Supplementary Fig. 9b) and exhibit multiple aneuploidies (3 and 4 aneuploid chromosomes while the average number of aneuploidies by line in this cross is of 2.23 aneuploidy) (Supplementary Figs. 5 and 6). Tetraploids show nonsignificant differences with diploids in LOH (most likely resulting from mitotic recombination) rate (Fig. 4c). However, they exhibit a higher number of the short-range LOH segments (two-sided Mann–Whitney–Wilcoxon test, $P = 5.4e-04$, Fig. 4d), indicating substantial differences in DNA repair process. It is worth noting that the LOH rate in tetraploids could be underestimated. In tetraploids, recombination events can only be detected between homeologous chromosomes, while recombination events between homologous chromosomes are undetectable. Consistent with this hypothesis, LOH analysis in triploid L1 and L2 hybrids shows a decreased LOH frequency leading to $SpC$ parent allele loss compared to diploids (Supplementary Fig. 14). The presence of two identical copies of chromosomes in allopolyploids may increase the efficiency of homologous recombination, thus improving the DNA repair process during mitosis and consequently could contribute to genome stabilization. Overall, neo-polyploid hybrids are characterized by increased genomic variability that can affect fitness and suggest that WGD might be a springboard for hybrid genome stabilization in the long term.

## Discussion

To investigate whether hybridization could increase the probability of WGD, we measured the rate of WGD in yeast intra-lineage crosses and hybrids using MA. We propagated hundreds of parallel crosses with serial bottlenecks to measure the neutral rate of genome doubling. We eliminated the potential confounding effect that natural selection could have on the distribution of traits correlated to fitness and genome dynamics, and consequently on genome duplication rate. Our results show that WGD occurs spontaneously in yeast intra-lineage crosses and hybrids within 90 cell divisions after mating, which may correspond to as early as 60 days in nature[55]. We find that some lineages are more prone to WGD, and that WGD can be triggered by hybridization, potentially through increased genomic instability. Our results indeed show that all crosses showing a higher rate of WGD exhibit more, although different, hallmarks of GIN compared to those where WGD did not occur.

WGD could have an impact on fertility and fitness that might in the long term influence the success of establishment of autopolyploids and allopolyploids as species. We found, as previous

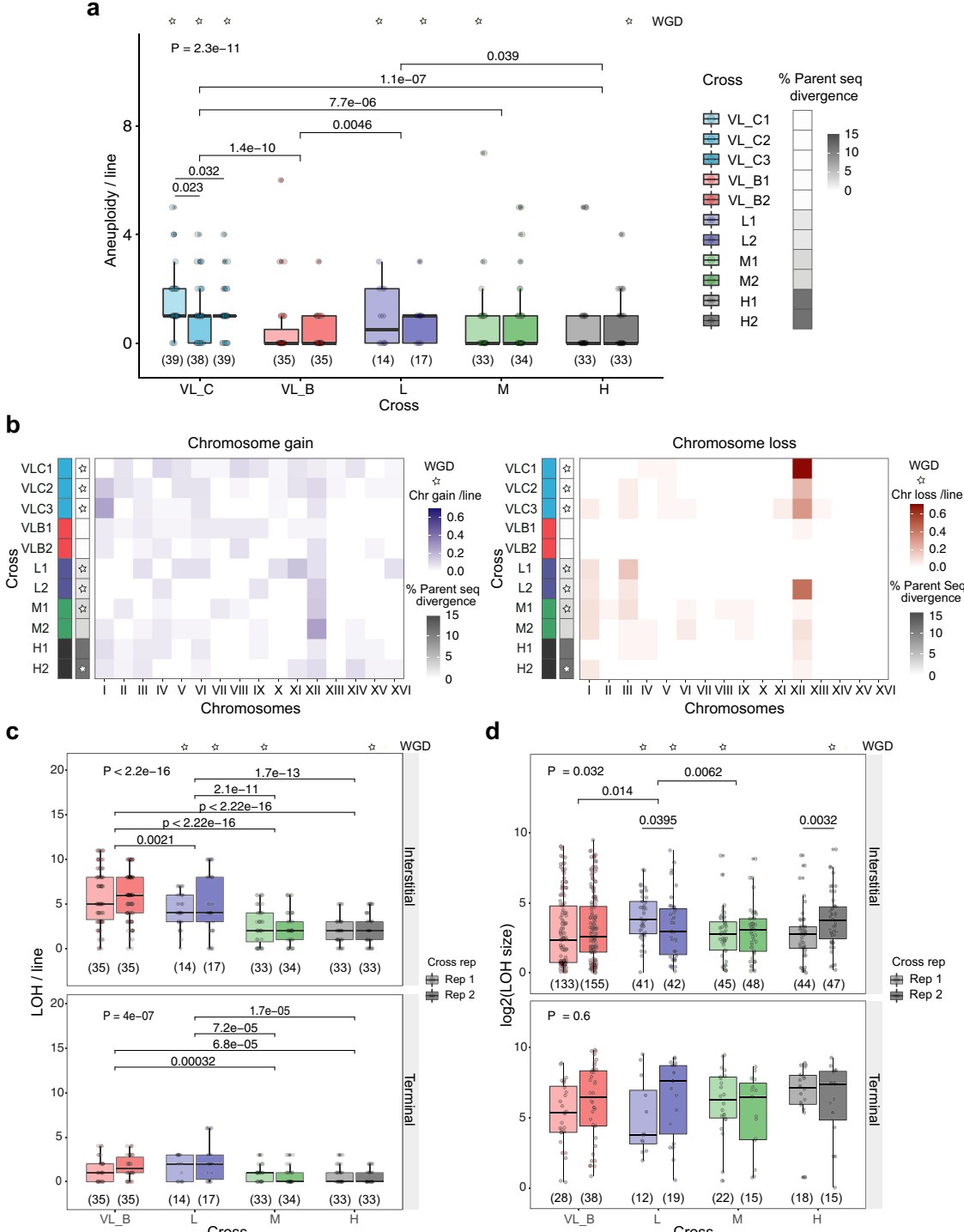

**Fig. 3 Higher rate of whole-genome duplication correlates with higher genomic instability. a** Aneuploidy rate among diploid intra-lineage crosses and hybrids. **b** Different patterns in chromosome gain (left panel) and loss (right panel) in diploid intra-lineage crosses and hybrids. **c** Loss of heterozygosity (LOH) rate in diploid intra-lineage crosses and hybrids decreases with genetic divergence ($P < 2.2e−16$ for interstitial LOH and $P = 4e−7$ for terminal LOH, Kruskal Wallis test). **d** Interstitial and terminal LOH segments size in diploid intra-lineage crosses and hybrids. The crosses with a star are those where whole-genome duplication (WGD) occurred. Numbers in parentheses represent the number of biologically independent lines (**a** and **c**) or independent LOH segments (**d**). *P* values from the Kruskal Wallis test (above) and a pairwise two-sided Mann–Whitney–Wilcoxon test are shown (only *P* values <0.05 are shown). For all boxplots the bold center line corresponds to the median value, the box boundaries correspond to the 25th and the 75th percentile, the whiskers correspond to 1.5 times the interquartile range, minimum and maximum values correspond to the minima and maxima and the dots correspond to the individual data points.

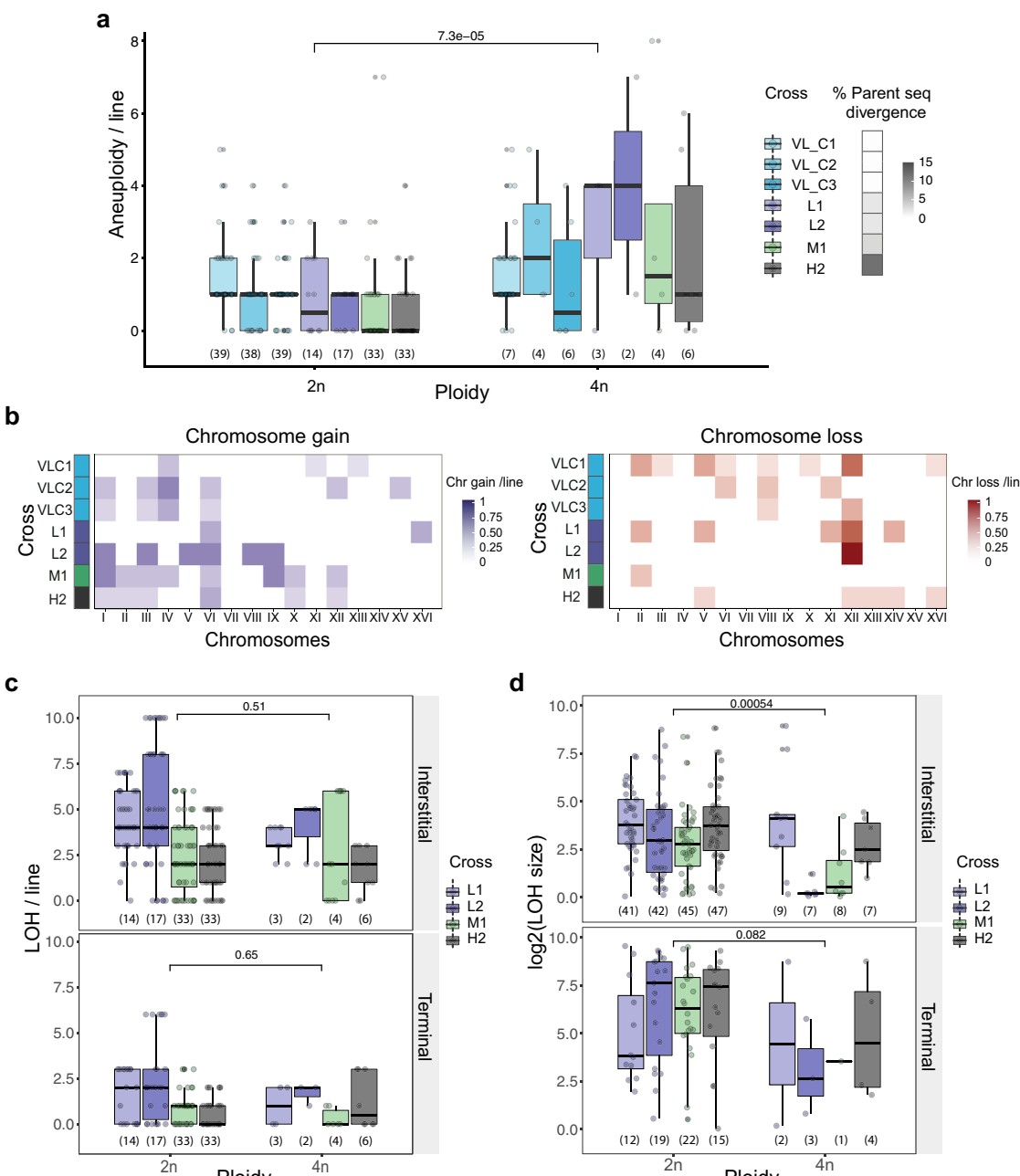

**Fig. 4 Tetraploid lines display increased genomic variability compared to diploids. a** Aneuploidy frequency increases with ploidy in intra-lineage VL_C crosses and hybrids. **b** Patterns of chromosome gain (left panel) and loss (right panel) in tetraploid intra-lineage VL_C crosses and hybrids. **c** Loss of heterozygosity (LOH) rate in tetraploids compared to diploid hybrids. **d** Interstitial and terminal LOH segment sizes in tetraploids compared to diploid hybrids. *P* values from a two-sided Mann–Whitney–Wilcoxon test are shown. Numbers in parentheses represent the number of biologically independent lines (**a** and **c**) or independent LOH segments (**d**). For all boxplots the bold center line corresponds to the median value, the box boundaries correspond to the 25th and the 75th percentile, the whiskers correspond to 1.5 times the interquartile range, minimum and maximum values correspond to the minima and maxima and the dots correspond to the individual data points.

studies did[51,52], that WGD systematically increases spore viability of yeast hybrids. Previous studies in *Saccharomyces* hybrids revealed that these species are isolated by a postzygotic double sterility barrier[26,76]. The first barrier is due to the failure of homeologous chromosomes pairing during meiosis. This barrier is overcome by WGD because it restores the correct pairing between identical homologous chromosomes. However, the generated viable spores are heterozygous at the mating-type (MAT) locus, which makes them unable to mate and thus causes the second sterility barrier. While the former sterility barrier is

analogous to the major mechanism of postzygotic reproductive isolation in plants and animals, the latter seems to be *Saccharomyces* specific[26,76].

The double sterility breakdown in yeast hybrids could occur through two possible paths. The first one involves WGD by endoreduplication, which is a consequence of cytokinesis failure, followed by the LOH in the MAT locus in the corresponding diploid spores. The resulting spores would produce vegetative cells capable of MAT switching and mating to produce a fertile allopolyploid. This path to fertility recovery was recently observed

in artificial *Saccharomyces* hybrids[76]. The double sterility break-down could also occur by means of damage to one copy of the MAT locus in the diploid hybrid. This damage to the MAT locus could cause hybrid cells to behave as a haploid, switch MAT and autotetraploidize. This path to fertility recovery was recently described in natural hybrid species of the *Zygosaccharomyces* genus[77,78]. However, in our experiment, MAT switching may not occur using the standard process because the necessary HO gene was deleted. The main way by which autotetraploidization could occur by mating in our study would be to have two hybrids with damage to the opposite MAT loci that are in the same colony and are close enough to mate with each other. Our data shows that this latter process is very unlikely since we did not detect genetic variations in chromosome III leading to such event and endor-eduplication is the most likely mechanism of WGD in our experiment[52].

Contrary to the prediction of Darlington[54], WGD has no negative effect on spore viability of intra-lineage crosses, which remains very high and similar to diploid fertility after WGD. Our results, in agreement with what was observed in some flowering plants[60], suggest that although four identical copies of chromosomes are present in individual yeast cells, pairing and segregation seems to occur correctly during meiosis in autopolyploid yeasts. However, WGD would create a second sterility barrier in half of the corresponding spores which will have heterozygous MAT loci. These diploid spores will however be able to sporulate and generate haploid spores competent for mating. Furthermore, tetraploids show a larger number of aneuploidies compared to diploids at the end of our experiment, which in the long term could decrease the spore viability of autotetraploids.

Our results show that spontaneous WGD leads to both gains and losses of mitotic fitness, such that WGD has no systematic trend toward fitness gain or loss in neutral conditions in both intra-lineage crosses and hybrids. It was previously shown that there is a general mitotic growth advantage for diploidy compared to other ploidies in yeast[21,79]. However, WGD can be advantageous under some stressful conditions[40,80]. Polyploids were shown to undergo significantly faster adaptation than their corresponding diploids, presumably due to an increased genomic instability leading to an increased phenotypic diversity[40]. Aneuploidy, concerted chromosome loss, and point mutations have been shown to provide large fitness gains in auto-polyploids evolved under nonpreferred carbon-source conditions[40]. The gains and losses of fitness observed after WGD in our experiment could be explained by beneficial or deleterious mutations accumulated during the neutral evolution of the independent tetraploid lines rather than by the direct effect of WGD. A recent study showed indeed that increased ploidy in euploid *S. cerevisiae* strains did not significantly affect growth rate[74]. Furthermore, we identified two allopolyploid lines that show a markedly increased and decreased fitness and exhibit multiple aneuploidies, supporting the implication of polyploidy in increasing genetic and phenotypic diversity.

Our data show that frequencies of aneuploidy in tetraploids are significantly higher than in diploids under neutral evolution and the frequency of chromosome loss is increasingly prevalent as ploidy increases. The same results were observed by comparing diploid and autopolyploid *S. cerevisiae* strains from anthropic populations[21,74] and experimental evolution under nonpreferred carbon-source conditions[40]. Our results suggest that these observations are not only the consequence of natural selection but also of the increased GIN in polyploids.

Interestingly, we find different patterns of chromosome loss in tetraploids compared to diploids. Chromosome loss, consistent with what was observed in *S. cerevisiae* diploids[21,62,73,74] (for review[75]) and meiotic products of intra-specific *S. paradoxus*

crosses[81], more frequently affects the smallest chromosomes (I and III) in diploid intra-lineage crosses and hybrids, probably because it affects a smaller number of genes. However, losses of larger chromosomes are observed in tetraploids, suggesting a less deleterious effect of their loss due to the multiple copies present in the cell and a lowered effect of genome imbalance compared to a large chromosome loss in diploids. Furthermore, we find that one of the allopolyploids (H2 hybrid) shows progressive reduction in ploidy from $4n$ to about $2.8n$ by large chromosome losses. Interestingly, this takes place from the same parental *S. paradoxus* subgenome. This strain also shows the most markedly increased growth rate among tetraploid lines, suggesting that these large chromosome losses may provide fitness gain to hybrids. Ploidy reversion by chromosome loss was previously observed in yeast autotetraploids[79] and allopolyploids[22,26,36,82]. The exact mechanism by which this reduction occurs in autopolyploids is unknown[1]. However, a recent study showed that the chromosome loss from one of the parental subgenomes could be the consequence of cytonuclear incompatibilities in an allopolyploid frog[83,84]. Increased GIN in polyploids could lead to ploidy reversion by chromosome loss as a way for genome stabilization on the long term[22,26].

Overall, our results show that GIN, whether intrinsic to some genotypes or triggered by hybridization, may accelerate WGD which in turn leads to increased fertility, genetic, and phenotypic diversity that may contribute to the establishment of both autopolyploids and allopolyploids. There is therefore a positive feedback between hybridization and WGD in evolution, hybridization could increase the rate of WGD, which in turn leads to more instability that fuels further adaptation.

## Methods

**Experimental crosses**. A total of 1304 lines from 15 different crosses were used; 864 of the lines from VL_B1/2, VL_C1/2/3, L1/2, M1/2, and H1/2 crosses and 152 of the lines from the VL_S2 cross were described previously[52,57,58]. In this study, we generated 288 new lines (3 new crosses): VL_B3 (96 lines), VL_A (96 lines) and VL_S1 (96 lines). These were generated using the same procedure as for the previously described lines[52]. Haploids were precultured overnight in 5 mL of YPD (1% yeast extract, 2% tryptone and 2% D-glucose). All incubation steps were performed at room temperature (RT). Precultures were then diluted at $OD_{600nm}$ of 1.0 in 500 μL aliquots. The aliquots from two strains to be crossed were mixed together and 5 μL were used to inoculate 200 μL of fresh YPD medium in 96 replicates so all individual strains derived from independent mating events and would be truly independent hybrids. Cells were given 6 h to mate after which 5 μL of the mating cultures were spotted on a diploid selection medium (YPD, 100 μg mL$^{-1}$ G418, 10 μg mL$^{-1}$ Nourseothricin). From each of the 96 spots, one colony was picked as a founding line for the evolution experiment, resulting in 96 independent lines for VL_B3, VL_A and VL_S1.

**Evolution experiment**. The 288 lines generated were evolved in the same conditions as the previously described lines by Charron et al.[52]. Each of the independent 288 lines (single colonies) was streaked on a sector corresponding to one-third of a YPD agar plate. Plates were incubated at RT for 3 days after which a new single colony was streaked as a progenitor for the new generation. The criteria for the new colony were to be (1) the closest to a predesigned mark on the Petri dish, allowing for unbiased colony selection, (2) a single colony, and (3) big enough to allow for both replication on a new medium and the inoculation of a liquid culture to generate a frozen stock. Every three passages, the colonies were both streaked and used to inoculate the wells of a 96 wells plate containing 150 μL of fresh YPD medium. After a 24 h of incubation at RT, 75 μL of 80% glycerol was added and the plates were placed in a −80 °C freezer for archiving. The lines were maintained on plates for a total of 35 passages, which is about 770 generations[52].

**Determination of ploidy**. Measurement of the cell DNA content was performed using flow cytometry with the SYTOX™ green staining assay (Thermo Fisher, Waltham, USA)[52]. Cells were first thawed from glycerol stocks on solid YPD in omnitray plates (RT, 3 days) including controls. The parental strains *S. paradoxus* SpB strain (MSH604) and the *S. cerevisiae* strain (LL13_054) were used as controls in both their haploid and diploid (wild strain) state. Liquid YPD cultures of 1 ml in 96-deep-well (2 ml) plates were inoculated and incubated for 24 h at RT. Cells were subsequently prepared as in Gerstein et al.[79], cells were first fixed in 70% ethanol for at least 1 h at RT. RNAs were eliminated from fixed cells using 0.25 mg/ml of

RNAse A during an overnight incubation at 37 °C. Cells were subsequently washed twice using sodium citrate (50 mM, pH7) and stained with a final SYTOX™ green concentration of 0.6 μM for a minimum of 1 h at room temperature in the dark. The volume of cells was adjusted to be around a cell concentration of less than 500 cells μL$^{-1}$. Five thousand cells of each sample were analyzed on a Guava® easyCyte 8HT flow cytometer using a sample tray for 96-well microplates. Cells were excited with the blue laser at 488 nm and fluorescence was collected with a green fluorescence detection channel (peak at 512 nm) using InCyte Software for Guava® easyCyte HT Systems (Ploidy data are deposited in figshare https://doi.org/10.6084/m9.figshare.12824258). The distributions of the green fluorescence values were processed to find the two main density peaks, which correspond to the two cell populations, respectively, in G1 and G2 phases (Supplementary Fig. 15). The data were analyzed using R version 4.0.4. The code is available at[85].

**Measurement of fertility**. Strains were thawed and 2 μL of the stocks were spotted on a fresh YPD medium and incubated for 3 days. A small number of cells were used to inoculate 4 mL of fresh YPD media and incubated for another day. From those precultures, a new 4 mL culture was inoculated at OD$_{600nm}$ of 0.6 in fresh YPD and grown for 3 h. Cells were subsequently prepared as in Charron et al.[52], cell cultures were centrifuged, the YPD was replaced with 4 mL of YEPA medium (1% yeast extract, 2% tryptone, and 2% potassium acetate). Cultures were incubated for 24 h after which they were centrifuged again, washed once with sterile deionized water, and put into 4 mL of SP medium (0.3% potassium acetate 0.02% d-Raffinose). After 5–7 days of incubation, the strains were dissected with a SporePlay™ dissection microscope (Singer Instruments, Somerset, UK) on YPD plates and incubated for 5 days at 25 °C. Pictures of the plates were taken after the incubation time and fertility was determined as the number of spores forming a colony visible to the naked eye after 5 days (Supplementary Data 1).

**Growth rate measurement**. A total of 233 strains (32 tetraploids at 4 timepoints and 35 diploids randomly selected at 3 timepoints) were thawed from glycerol stocks on solid YPD omnitray plates (25 °C, 72 h). Four to five independent replicates from each strain were precultured in 1 mL of YPD liquid cultures in 96 deep-well plates (2 ml) and incubated for 24 h at 25 °C. Precultures were then diluted to OD$_{600nm}$ = 0.1 and incubated at 25 °C to reach the OD$_{600nm}$ = 0.6. Subsequently, 20 μL of these precultures were grown in 96-well flat-bottomed culture plates in 180 μL of media (YPD), resulting in an initial OD$_{600nm}$ of approximately 0.1. Incubation at 25 °C was performed directly in three temperature-controlled spectrophotometers (Infinite® 200 PRO, Tecan, Reading, UK) that read the OD$_{600nm}$ at intervals of 15 min using the MagellanTM data analysis software (Supplementary Data 2). The growth rate of each replicate was estimated from growth curves using R v4.0.4 (the code is available at[85]). The growth rate was computed as the 98th percentile of the set of linear regression slopes fitted in ten-timepoint wide overlapping sliding windows.

**Whole-genome sequencing**. We performed whole-genome sequencing of 895 strains, 757 were also reported in[57]:

1. 33 to 40 randomly selected diploid lines from VL_B1, VL_B2, VL_C1, VL_C2, VL_C3, M1, M2, H1, H2;
2. 13 and 16 diploid and 19 and 17 triploid lines from L1 and L2;
3. all of the 32 strains that were tetraploid were sequenced soon after mating and at the end of the experiment (and in the middle of the experiment after 352 generations for tetraploids that show ploidy change to diploid);
4. the 13 corresponding haploid parental strains.

Genomic DNA was extracted from overnight cultures from one isolated colony of each stock following standard protocols (QIAGEN DNAeasy, Hilden, Germany). Extracted DNA was treated with RNase A and purified on Axygen AxyPrep Mag PCR Clean-up SPRI beads. Libraries were prepared with the Illumina Nextera kit (Illumina, San Diego, USA) following the manufacturer's protocol and modifications from[86]. The quality of a few randomly selected libraries was controlled using an Agilent BioAnalyzer 2100 electrophoresis system. Pooled libraries were sequenced in paired-end, 150 bp mode on different lanes of HiSeqX (Illumina, San Diego, USA) at the Genome Quebec Innovation Center (Montréal, Canada). The 895 genomes were sequenced with an average genome-wide coverage of 90×. Raw sequences are accessible at NCBI (bio project ID PRJNA515073).

**Read mapping**. Raw reads were trimmed using Trimmomatic version 0.33 with parameters ILLUMINACLIP:nextera.fa:6:20:10 MINLEN:40 and a library of Illumina Nextera adapter sequences. Reads were subsequently mapped on the reference genome of *S. paradoxus* MSH604 strain, one of the four used *SpB* parental strains, for VL_B crosses and all hybrids, onto the *SpC* parental strain (LL2011_012) for the VL_C crosses, and onto *S. cerevisiae* YPS128 strain for H hybrids using bwa mem v0.7.17. Mapped reads were sorted using samtools sort

version 1.8. and duplicates were removed using Picard RemoveDuplicates version 2.18.29-SNAPSHOT with parameter REMOVE_DUPLICATES = true.

**Read depth**. Read coverage for each position in the genome was estimated using SAMtools depth v1.8. and averaged over 10 kb windows to detect copy number variation within and among chromosomes. The read depth of coverage obtained for each bin of 10 kb on each chromosome was divided by the whole-genome coverage. The median of the values obtained for each chromosome corresponds to the chromosome copy number and the difference between these values at $T_{end}$ (at the end of the experiment) and $T_{ini}$ (soon after mating) corresponds to the number of gained or lost chromosomes. The data was analyzed using R version 4.0.4 (the code is available at[85]).

**Allele frequency and LOH**. Single nucleotide polymorphisms (SNP) were called using Freebayes v1.3.1[87] accounting for different ploidy levels of strains and chromosomes. Control-FREEC v11.5[88,89] was used to determine copy number in 250 bp nonoverlapping windows of each genome with options break-PointThreshold = 0.8, minExpectedGC = 0.35, maxExpectedGC = 0.55, telocentromeric = 7000. The most common copy number occurring across windows for a given chromosome in a given strain was set as the copy number of that strain and that chromosome. Variant calling was run for each cross replicate separately including both haploid parental strains. The following options were used: -q 20 --use-best-n-alleles 4 --limit-coverage 20000 -F 0.02. Multi-nucleotide polymorphisms were decomposed into SNPs using script vcfallelicprimitives with -kg option. The criteria for selecting SNPs included QUAL > 1, QUAL/AO > 10 (where QUAL is a quality of variant and AO is the number of alternate alleles), SAF > 0 (number of alternate observations on the forward strand), SAR > 0 (number of alternate observations on the reverse strand), RPR > 1 (number of reads centered to the right of an alternate allele), RPL > 1 (number of reads centered to the left of an alternate allele), MQM / MQMR > 0.9 and MQM / MQMR < 1.05 (where MQM and MQMR are mean mapping qualities of alternate and reference alleles, respectively). Indels, multiallelic SNPs, and SNPs overlapping repeats were excluded.

For allele frequency calculation, only heterozygous loci between parental genomes for each cross were kept and the parental origin of each allele was identified. Allele frequencies of heterozygous loci were calculated as the allele read depth corresponding to one of the two parents divided by the total read depth of the locus. Only loci with read depth higher than 20 reads were considered. LOH tracts were identified. SNPs with allele frequencies deviating from the average allele frequency over the whole chromosome (±0.15) were kept as potential SNPs belonging to an LOH tract. Stretches of consecutive marker positions were grouped in LOH regions. Blocks of LOH were identified by looking for a minimum of three successive SNPs with the same allele frequency (±0.1) in a window of 300 bp for hybrids and a window of 1 kb for VL_B crosses. The size of LOH segments was calculated as the difference between the position of the last and the first SNP of each identified LOH segment. Because the minimum LOH size was of 1 kb in VL_B crosses due to the very low heterozygosity in these crosses, we considered in our comparisons of LOH frequencies between crosses only LOH segments larger than 1 kb. The very low heterozygosity in VL_C crosses prevent the accurate measurement of the LOH rate in these crosses. The data were analyzed using R version 4.0.4 (the code is available at https://github.com/Landrylab/Marsit_et_al_2021[85]).

**Statistical analyses**. Statistical analyses and figure creation for all data were done using custom scripts available at https://github.com/Landrylab/Marsit_et_al_2021[85] in R version 4.0.4.

**Reporting summary**. Further information on research design is available in the Nature Research Reporting Summary linked to this article.

## Data availability

The raw sequencing data generated in this study have been deposited in the Sequence Read Archive (SRA), NCBI database under accession code PRJNA515073. The raw flow cytometry data generated in this study have been deposited in figshare with the identifier [https://doi.org/10.6084/m9.figshare.12824258]. The fertility data and raw optical density for growth rate analysis generated in this study are provided in Supplementary Data 1 and 2. The remaining data are available within the article or from the authors upon request. Source data are provided with this paper.

## Code availability

Custom scripts available at Landrylab/Marsit_et_al_2021 [https://doi.org/10.5281/zenodo.4633600] in R version 4.0.4.

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

## Acknowledgements

We thank H. Martin for contributing to preliminary sequence data analyses and D.W. Hall for providing yeast MA lines. We also thank the members of the Landry lab for discussions and M. Barker, A.M. Dion-côté, C. Bautista Rodriguez, A. F. Cisneros, and D. Ascencio, for comments on the manuscript. NSERC Discovery grant and Canada Research Chair to C.R.L, FRQS post-doctoral fellowship to S.M, FRQNT scholarship to M.H and G.C, NSERC Alexander Graham-Bell scholarship to M.H and G.C.

## Author contributions

Conceptualization, C.R.L. and S.M.; Data curation, S.M. and M.H; Funding acquisition, C.R.L.; Experimental work: S.M., M.H., and G.C, Formal analysis: S.M. and A.F. Project supervision, C.R.L.; Writing - original draft, S.M and C.R.L.; Writing – review & editing, all authors.

## Competing interests

The authors declare no competing interests.
