## [Peer Review File · Nature Communications]

REVIEWER COMMENTS

Reviewer #1 (Remarks to the Author):

In this manuscript, the authors explore the link between whole-genome duplication and hybridization in *Saccharomyces* yeast. The authors use a well-established and well-studied model system for hybridization research, employing different populations with *S. paradoxus* for intraspecific hybridization and *S. paradoxus* x *S. cerevisiae* interspecific hybrids. The methods used are solid and robust and have been implemented to answer a long-standing question in the field of hybridization, is WGD a direct result of hybridization or a coping mechanism. The analyses are complex, with multiple replicates and varying degrees of genomic divergence (VL, L, M, H). However, I feel that the authors were able to succinctly report their findings in an understandable fashion, despite the complexity. Overall, I think the paper is well-written and the scientific conclusions are well founded and appropriately reported. I found no major issues or comments.

I do have a few minor/figure comments.

Minor comments:

- (1) Line 114: On first reading, I was slightly confused of why there was a range 0 to 11% that went through WGD. You might want to re-word this sentence slightly for clarity.
- (2) Line 123-125: Due to spontaneous diploidization of strains from SpC, almost half of the L1 and L2 lines are triploid and were triploid at the initial timepoint (Fig. S1). These same SpC parents were then also used in the VL_C crosses with SpB. Can the authors explain why we don't see any triploid in the VL_C crosses at the initial timepoint? If the SpC parents are so prone to spontaneous diploidization, I would expect to see a few triploids.
- (3) Line 139-140: Any hypothesis for extinction of two tetraploids during the experiment or just random chance?
- (4) Line 166: Is H2_43 shown in Fig. 2B? Or due to loss of sporulation is there no fertility reported for this hybrid? After reading the methods more thoroughly I now see that fertility is calculated as spore viability. I would suggest adding a brief note on how fertility was calculated either in the manuscript text or the figure legend for Fig. 2B.
- (5) Line 530: I believe it should read "all of the 32 strains that were tetraploid..."

Figure comments:

- (1) It might be helpful to add Gain and Loss titles to Fig. 3B and Fig. 4B.
- (2) The order of supplemental figures seems to not follow the order in which they are referenced in the figure.

Reviewer #2 (Remarks to the Author):

The submitted manuscript reports on the investigation of genome duplication in *Saccharomyces* hybrids. The analysis of a large number of intra- and interspecies hybrids revealed a correlation between the genomic divergence of the partners and the rate of genome duplication in their hybrids. It is also shown that the hybrid genomes are unstable and segregate during prolonged cultivation. Interestingly, the authors use yeasts to study these processes, but ignore most of the rather large amount of knowledge accumulated about interspecies *Saccharomyces* hybrids over the past decade. Because of this, some of the findings appear as new discoveries in the manuscript, although they actually only confirm previous results from other laboratories. When interpreting their data, authors should take into account previous findings and explain where and why they agree / disagree with the views of other authors. More specific comments:

1. Numerous previous papers reported on observations indicating that interspecies *Saccharomyces* hybrids are prone to double their genome size. Thus, this manuscript has a misleading title: the phenomenon is already known.
2. The authors do not compare/confront their results with those published before their work. By comparing their findings with the results of other studies, they could better highlight the novelty in

their work.

3. The statements that „WGD can restore fertility“ „WGD increases fitness by restoring fertility“ (e.g. lines 84, 160) need to be revised. Several papers have shown that genome duplication is necessary but not sufficient for fertility restoration in *Saccharomyces* interspecies hybrids (in contrast to plant hybrids) because the gametes (ascospores) produced by the yeast allotetraploids are usually sterile (do not function as gametes, cannot fertilise/mate). This is an essential difference from the plant allotetraploids, whose allodiploid gametes are active. This difference is due to the different mode of sex determination. The allodiploid yeast gamete is heterozygous at the MAT locus; MATa/MATalpha heterozygosity represses the mating-specific genes required for fertilisation. See the literature for more detail in *Saccharomyces* and *Zygosaccharomyces* interspecies hybrids (e.g. doi: 10.1093/femsyr/foy079; <https://doi.org/10.1007/s00294-020-01080-0>). The authors seem to overlook this difference between the plant and yeast systems.

4. The method used for measuring fertility is incomplete. It only measures the viability of spores (gametes). It does not test them for fertility (ability to mate with other gametes). Viable gametes are not automatically functional (see previous and next comments).

5. The observed instability of large chromosomes may be (parental) strain-specific because other works have reported on high instability of small chromosomes. The instability of Chr III is especially important because its malsegregation at meiosis can break down the MAT heterozygosity. Loss of MAT heterozygosity makes the gametes (ascospores) of the allotetraploid fertile. Again, the comparison of the authors' own observations (unstable large chromosomes) with the results published in previous works (unstable small chromosomes) could strengthen the paper.

6. Postzygotic instability of genomes is also a known phenomenon in yeast interspecies hybrids. In spite of this, the authors refer in line 196 to hardly relevant papers on cancer cells which are neither yeast cells nor allopolyploid hybrids.

7. Many previous papers reported on the genetic instability and segregation of allotetraploids. Chromosomes can be lost (usually and preferentially from one of the subgenomes) both at mitotic and at meiotic divisions resulting in broad spectra of segregants of chimeric (mosaic) genomes. The authors' results described in lines 141-150 are consistent with the previous observations which should be mentioned. It would be even more useful to compare the authors' findings with previous ones to show how their results contribute to the better understanding of the postzygotic evolution of the hybrid genomes.

8. Preferential pairing of chromosomes within the subgenomes was recently investigated in *Saccharomyces* interspecies hybrids. The phenomenon is called allosyndetic pairing (or autodiploidisation of the allotetraploid meiosis). The terms are borrowed from plant genetics, but in contrast to plants, it does not restore fertility but maintains the sterility by producing non-fertilising MATa/MATalpha allodiploid gametes. Thus, we know much more about it than described as a hypothesis in the (almost one-hundred-year old) reference cited in the lines 151-159.

9. The terms di-, tetra- and aneuploid should be used consistently and clearly. E.g. if a tetraploid hybrid loses a chromosome or a pair of homologous chromosomes, it is no longer tetraploid. It is a trisomic or a disomic aneuploid segregant/derivative of the allotetraploid hybrid.

10. Please be more specific about LOH. Loss of heterozygosity can be due to the loss of an entire chromosome (or a pair of homologous chromosomes in allotetraploids) from one subgenome or to (the much less likely) gene conversion/recombination between allosyndetic (homeologous) chromosomes of the subgenomes. Is there any correlation between the loss of chromosomes and the loss of heterozygosity at genes located on those chromosomes?

11. Timing of genome duplication. Lines 137-138: „Genome doubling occurred at different generation time points after mating; some of them emerged quickly, in less than 90 generations, while others appeared after more than 680 generations“. Wouldn't be it more accurate to say that the number of tetraploid cells reached the threshold of detectability after XX generations? Can you exclude the possibility that the genome-doubling event or events took place much earlier than the e.g. 90th generation and became detectable much later, only after they had grown to a detectable subpopulation (by e.g. the 90th generation) (provided they were competitive!!!).

12. Line 114 „within 770 generations, 0 to 11% of the lines went through spontaneous WGD.“ This formulation is somewhat confusing. Do the authors want to say that 11 % of the diploid hybrid cells converted to tetraploidy (millions of independent genome doubling events) by the time of examination or they mean that 11% of the population was tetraploid at the time of examination. I doubt that the former can be measured. If the latter was meant by the authors then they cannot say that the percentages of tetraploids reflect the frequency of (disposition to) genome duplication

because even a high percentage can be the result of a single duplication event if the allotetraploid is more competitive than the allodiploid in the applied culturing conditions. Its percentage may simply reflect the time point of the event: the higher the percentage, the earlier the event.

13. What is the difference between the pseudohaploid and the diploid cell (line 124)? If the genome is diploid (due to e.g. endomitosis), the cell is diploid regardless of the homo-/heterozygosity at the MAT locus. To call it pseudohaploid may cause confusion. Mating competence is not a privilege of haploids.

14. Etc.

15. Lists of publications relevant for the comments can be found in recent reviews: e.g. doi.org/10.1002/yea.3294, [10.3389/fmicb.2018.03071](https://doi.org/10.3389/fmicb.2018.03071), [10.1093/femsyr/foaa040](https://doi.org/10.1093/femsyr/foaa040)

REVIEWER COMMENTS

Reviewer #1 (Remarks to the Author):

1.1 In this manuscript, the authors explore the link between whole-genome duplication and hybridization in *Saccharomyces* yeast. The authors use a well-established and well-studied model system for hybridization research, employing different populations with *S. paradoxus* for intraspecific hybridization and *S. paradoxus* x *S. cerevisiae* interspecific hybrids. The methods used are solid and robust and have been implemented to answer a long-standing question in the field of hybridization, is WGD a direct result of hybridization or a coping mechanism. The analyses are complex, with multiple replicates and varying degrees of genomic divergence (VL, L, M, H). However, I feel that the authors were able to succinctly report their findings in an understandable fashion, despite the complexity. Overall, I think the paper is well-written and the scientific conclusions are well founded and appropriately reported. I found no major issues or comments.

Answer 1.1: We thank [reviewer's name has been removed to maintain confidentiality] for the overall very positive evaluation of our work.

I do have a few minor/figure comments.

Minor comments:

1.2 Line 114: On first reading, I was slightly confused of why there was a range 0 to 11% that went through WGD. You might want to re-word this sentence slightly for clarity.

Answer 1.2: We changed the sentence and it now reads as (line 127-128):

“This shows that within 770 generations, spontaneous WGD occurred in 0 to 11% of the populations among the different crosses.”

1.3 Line 123-125: Due to spontaneous diploidization of strains from SpC, almost half of the L1 and L2 lines are triploid and were triploid at the initial timepoint (Fig. S1). These same SpC parents were then also used in the VL_C crosses with SpB. Can the authors explain why we don't see any triploid in the VL_C crosses at the initial timepoint? If the SpC parents are so prone to spontaneous diploidization, I would expect to see a few triploids.

Answer 1.3: We agree with the reviewer, however, we previously showed that the proportion of diploid cells in the parental haploid stock appears to be quite variable and is consistent with stochastic switching at low frequency (Charron, Marsit et al., 2019). We previously repeated the experiment of ploidy analysis of the parental stock twice, the first time no diploid was identified among 94 tested isolated colonies. However, the second time 5 among 94 were identified as diploid (Charron, Marsit et al., 2019). This suggests that the duplication of haploid cells in the parental stock is a stochastic event. Moreover, the frequency of triploids in L1 and L2 crosses is variable and may vary from 1% to 57% as previously shown (Charron, Marsit et al., 2019). These results suggest that the proportion of triploids might depend on the initial proportion of diploidized parents (competent for mating) in the preculture, which itself is stochastic. The frequency of diploids in the preculture used for crosses might also depend on at which phase of cell growth it appears to be more or less widespread in the cell culture.

1.4 Line 139-140: Any hypothesis for extinction of two tetraploids during the experiment or just random chance?

Answer 1.4: We believe that the number of the extinct lines is too low to draw strong conclusions from these observations. However, we can speculate. The extinction of the two tetraploids L1_31 and H2_38 before the end of the experiment could be due to the emergence of deleterious changes that take place, mutations or aneuploidies. In our mutation accumulation experiment, not all lines could be propagated through 770 mitotic generations. We showed that 77.9% of the diploid lines were still propagated after 770 generations (Charron, Marsit et al., 2019). The diploid L and H hybrids show the lowest proportion of surviving lines (averages of 69.3% for L, 70.3% for H) compared to VL (average of 95.8%). The data suggest that the line extinction may be the consequence of the frequent segregation of highly deleterious variants generated by genome instability. Among the 32 tetraploids, only two declined and they belong to L and H hybrids for which diploids show the lowest proportion of surviving lines, suggesting that these hybrids remain relatively more unstable than the other crosses after WGD.

1.5 Line 166: Is H2_43 shown in Fig. 2B? Or due to loss of sporulation is there no fertility reported for this hybrid? After reading the methods more thoroughly I now see that fertility is calculated as spore viability. I would suggest adding a brief note on how fertility was calculated either in the manuscript text or the figure legend for Fig. 2B.

Answer 1.5: The tetraploid line (H2_43) is not shown in Fig. 2B because indeed this line lost its ability to sporulate and in this figure fertility corresponds to the percentage of viable spores. We now added a brief note on how fertility was calculated in the manuscript text (line 179-180):

“One typical proxy for fertility in yeast is the percentage of viable spores after meiosis⁵³.”

Fig. 2B legend (line 788) and the Supplementary Figure 7:

“Fertility here is measured using as proxy the percentage of viable spores after meiosis.”

1.6 Line 530: I believe it should read “all of the 32 strains that were tetraploid...”

Answer 1.6: We now corrected this sentence (line 477):

“all of the 32 strains that were tetraploid were sequenced”

Figure comments:

1.7 It might be helpful to add Gain and Loss titles to Fig. 3B and Fig. 4B.

Answer 1.7: We now added “Chromosome gain” and “Chromosome Loss” titles to Fig. 3B

and Fig. 4B

1.8 The order of supplemental figures seems to not follow the order in which they are referenced in the figure.

Answer 1.8: We now corrected the order of supplemental figures.

Devin Bendixsen

Reviewer #2 (Remarks to the Author):

The submitted manuscript reports on the investigation of genome duplication in *Saccharomyces* hybrids. The analysis of a large number of intra- and interspecies hybrids revealed a correlation between the genomic divergence of the partners and the rate of genome duplication in their hybrids. It is also shown that the hybrid genomes are unstable and segregate during prolonged cultivation.

2.1 Interestingly, the authors use yeasts to study these processes, but ignore most of the rather large amount of knowledge accumulated about interpecies *Saccharomyces* hybrids over the past decade. Because of this, some of the findings appear as new discoveries in the manuscript, although they actually only confirm previous results from other laboratories.

Answer 2.1: We are sorry if it was not clear that we acknowledge the rich literature of yeast hybrids. We agree that much work has been done, including by our group. We have reviewed some of it in our recent review paper published in *Nature Reviews Genetics*. Our first version was prepared for *Nature* and transferred to *Nature Communications*, which means that it was a very short format. We now have more room for discussion. As you will see below, we now better acknowledge these previous studies. However, we are not aware of any studies that have tested whether hybrids were more prone to WGD than non-hybrids. The fact that hybrids are often tetraploids could come from the fact that they are much fitter as tetraploids, so we just see them, or that they are as fit as diploid but they have a high rate of WGD. This is exactly the same situation as it has been seen in plants, as we explain in our introduction. In fact, the impossibility to disentangle these two scenarios from observations in nature is the reason why we did these experiments in the first place. Testing this needs to compare, side by side, the rate of WGD in hybrids and parents, in a context in which natural selection plays little role so we can measure rates of WGD almost independent of its effect on long-term fitness. We are sorry if this did not come through clearly in the previous version of our manuscript. We now developed this more in the introduction (line 63-97):

*“Autopolyploid and allopolyploid yeasts have been described in anthropic and some natural environments¹³⁻²¹. In the budding yeast *Saccharomyces cerevisiae*, most of the natural isolates are diploid. However, polyploid isolates (3–5n) (11.5% of 1011 yeast genomes) are enriched among domesticated strains, suggesting that some human-related environments have had an effect on triggering changes to higher ploidy levels or favoring the propagation of polyploid strains¹⁸. Bakery, Ale beer and a fraction of clinical strains of *S. cerevisiae* are frequently auto-tetraploid¹⁴⁻¹⁷. Polyploidy is also common in recent yeast hybrids^{12,19-22}, suggesting that allopolyploidization might not be a rare mechanism of yeast hybridization¹². Many industrial strains are allopolyploids resulting from hybridization*

between different *Saccharomyces* species^{12,19–21}. The lager beer yeast *S. pastorianus* was shown to be an allopolyploid hybrid between an ancestor of the ale beer yeast *S. cerevisiae* and *S. eubayanus*²³. Allopolyploid wine and cider yeasts have also been reported, including *S. cerevisiae* × *S. kudriavzevii*, *S. cerevisiae* × *S. uvarum* and *S. cerevisiae* × *S. kudriavzevii* × *S. uvarum* hybrids^{24–29}. Compared with the parental strains, these hybrids have several advantages for wine and beer fermentation, including increased tolerance to various stresses and increased fermentative performance^{19–21,30,31}. Accordingly, only a few cases of non-synthetic *Saccharomyces* homoploid hybrids (same ploidy as parents) have been reported^{32,33}. While significant progress has been made in the last decade in understanding the consequences of WGD^{1,34}, the factors that favor its evolution are less well studied³⁵. A large body of work, mostly on plants, showed that hybrid speciation often coincides with WGD¹. A long standing hypothesis suggests that parental genetic divergence influences the probability of WGD in hybrids³⁶. Combining two diverged genomes in the same cells would increase the rate of genomic changes and WGD, which would in turn enable hybrid maintenance on the long term³⁷. Recent studies showed that homoploid hybrid plants tend to be derived from parents that are less evolutionarily divergent than parents of polyploid hybrids^{38–43}. However, the role of genetic divergence as a driver for WGD has been debated^{38–43}. The debate comes from the fact that the positive correlation between parental divergence and polyploidization could be related to the success of hybrid establishment rather than the rate of WGD itself. Allopolyploids could be more likely to be maintained because WGD in hybrids increases fitness by restoring fertility^{36,44–46} and accelerating adaptation⁴⁷. On the contrary, auto-polyploids frequently have reduced fertility due to multivalent pairing during meiosis⁴⁸. These effects combined would increase the frequency of WGD in hybrid species. Testing these alternative models, increased rates of WGD versus enhanced adaptive potential of hybrids following WGD, requires removing natural selection from the equation.”

When interpreting their data, authors should take into account previous findings and explain where and why they agree / disagree with the views of other authors. More specific comments:

2.2 Numerous previous papers reported on observations indicating that interspecies *Saccharomyces* hybrids are prone to double their genome size. Thus, this manuscript has a misleading title: the phenomenon is already known.

Answer 2.2: Our title does not claim that nothing was known on the subject before our work. We already cited papers from yeast that reported WGD (Marcet-Houben et al., 2015; Morales et al., 2012; Charron et al., 2019). As mentioned above, we do not know of any studies that have measured the rate of WGD to demonstrate experimentally that hybridization can trigger WGD although we do know that reports of hybrid tetraploid strains have been published.

We now added more details on hybrids and polyploids in yeast in the introduction (line 63-78):

“Autopolyploid and allopolyploid yeasts have been described in anthropic and some natural environments^{13–21}. In the budding yeast *Saccharomyces cerevisiae*, most of the natural isolates are diploid. However, polyploid isolates (3–5n) (11.5% of 1011 yeast genomes) are enriched among

domesticated strains, suggesting that some human-related environments have had an effect on triggering changes to higher ploidy levels or favoring the propagation of polyploid strains¹⁸. Bakery, Ale beer and a fraction of clinical strains of *S. cerevisiae* are frequently auto-tetraploid¹⁴⁻¹⁷. Polyploidy is also common in recent yeast hybrids^{12,19-22}, suggesting that allopolyploidization might not be a rare mechanism of yeast hybridization¹². Many industrial strains are allopolyploids resulting from hybridization between different *Saccharomyces* species^{12,19-21}. The lager beer yeast *S. pastorianus* was shown to be an allopolyploid hybrid between an ancestor of the ale beer yeast *S. cerevisiae* and *S. eubayanus*²³. Allopolyploid wine and cider yeasts have also been reported, including *S. cerevisiae* × *S. kudriavzevii*, *S. cerevisiae* × *S. uvarum* and *S. cerevisiae* × *S. kudriavzevii* × *S. uvarum* hybrids²⁴⁻²⁹. Compared with the parental strains, these hybrids have several advantages for wine and beer fermentation, including increased tolerance to various stresses and increased fermentative performance^{19-21,30,31}. Accordingly, only a few cases of non-synthetic *Saccharomyces* homoploid hybrids (same ploidy as parents) have been reported^{32,33}.

2.3 The authors do not compare/confront their results with those published before their work. By comparing their findings with the results of other studies, they could better highlight the novelty in their work.

Answer 2.3: We thank the reviewer for this suggestion. Because of the initial submission instructions, we were indeed limited in word numbers and references leading to short and condensed discussions. We now added in the discussion several paragraphs where we compare our findings with the results of other studies to better highlight the novelty of our work.

(Line 315-318) “We found, as previous studies did^{45,46}, that WGD systematically increases spore viability of yeast hybrids. Previous studies in *Saccharomyces* hybrids revealed that these species are isolated by a postzygotic double sterility barrier^{31,68}.”

(Line 343-346) “Our results, in agreement with what was observed in some flowering plants⁵⁴, suggest that although four identical copies of chromosomes are present in individual yeast cells, pairing and segregation seems to occur correctly during meiosis in auto-polyploid yeasts.”

(Line 361-365) “The gains and losses of fitness observed after WGD in our experiment could be explained by beneficial or deleterious mutations accumulated during the neutral evolution of the independent tetraploid lines rather than by the direct effect of WGD. Recent study showed indeed that increased ploidy in euploid *S. cerevisiae* strains did not significantly affect growth rate⁶⁶.”

(Line 371-374) “The same results were observed by comparing diploid and auto-polyploid *S. cerevisiae* strains from anthropic populations^{18,66} and experimental evolution under non preferred carbon-source conditions³⁴, suggesting that these observations are not only the consequence of natural selection but also of the increased GIN in polyploids.”

(Line 377-379) “Chromosome loss, consistent with what was observed in *S. cerevisiae* diploids^{18,56,66} (for review⁶⁷) and meiotic products of intra-specific *S. paradoxus* crosses⁷², more frequently affects the smallest chromosomes”

(Line 387-392) “Ploidy reversion by chromosome loss was previously observed in yeast auto-tetraploids⁷¹ and allopolyploids^{19,29,31,73}. The exact mechanism by which this reduction occurs in auto-polyploids is unknown¹. However, a recent study showed that the chromosome loss from one of the parental subgenomes could be the consequence of cytonuclear incompatibilities in an allopolyploid frog⁷⁴. Increased GIN in polyploids could lead to ploidy reversion by chromosome loss as a way for genome stabilisation^{19,31}.”

2.4 The statements that „WGD can restore fertility“ „WGD increases fitness by restoring fertility“ (e.g. lines 84, 160) need to be revised. Several papers have shown that genome duplication is necessary but not sufficient for fertility restoration in *Saccharomyces* interspecies hybrids (in contrast to plant hybrids) because the gametes (ascospores) produced by the yeast allotetraploids are usually sterile (do not function as gametes, cannot fertilise/mate). This is an essential difference from the plant allotetraploids, whose allodiploid gametes are active. This difference is due to the different mode of sex determination. The allodiploid yeast gamete is heterozygous at the MAT locus; MATa/MATalpha heterozygosity represses the mating-specific genes required for fertilisation. See the literature for more detail in *Saccharomyces* and *Zygosaccharomyces* interspecies hybrids (e.g. doi: 10.1093/femsyr/foy079; <https://doi.org/10.1007/s00294-020-01080-0>). The authors seem to overlook this difference between the plant and yeast systems.

Answer 2.4: We agree that spore survival is only one of many fitness components. We now add details on this in the discussion (line 314-340):

*“WGD could have an impact on fertility and fitness that might in the long term influence the success of establishment of auto-polyploids and allopolyploids as species. We found, as previous studies did^{45,46}, that WGD systematically increases spore viability of yeast hybrids. Previous studies in *Saccharomyces* hybrids revealed that these species are isolated by a postzygotic double sterility barrier^{31,68}. The first barrier is due to the failure of homeologous chromosomes pairing during meiosis. This barrier is overcome by WGD because it restores the correct pairing between identical homologous chromosomes. However, the generated viable spores are heterozygous at the mating type locus, which makes them unable to mate and thus causes the second sterility barrier. While the former sterility barrier is analogous to the major mechanism of postzygotic reproductive isolation in plants and animals, the latter seems to be *Saccharomyces*-specific^{31,68}.*

*The double sterility breakdown in yeast hybrids could occur through two possible paths. The first one involves WGD by endoreduplication, which is a consequence of cytokinesis failure, followed by the loss of heterozygosity in the MAT locus in the corresponding diploid spores. The resulting spores would produce vegetative cells capable of mating type switching and mating to produce a fertile allopolyploid. This path to fertility recovery was recently observed in artificial *Saccharomyces* hybrids⁶⁸. The double sterility breakdown could also occur by means of damage to one copy of the mating type (MAT) locus in the diploid hybrid. This damage to the MAT locus could cause hybrid cells to behave as a haploid, switch mating type and*

autotetraploidize. This path to fertility recovery was recently described in natural hybrid species of the Zygosaccharomyces genus^{69,70}. However, in our experiment, mating type switching may not occur using the standard process because the necessary HO gene was deleted. The main way by which autotetraploidization could occur by mating in our study would be to have two hybrids with damage to the opposite MAT loci that are in the same colony and are close enough to mate with each other. Our data shows that this latter process is very unlikely since we did not detect genetic variations in chromosome III leading to such event and endoreduplication is the most likely mechanism of WGD in our experiment⁴⁶.”

2.5 The method used for measuring fertility is incomplete. It only measures the viability of spores (gametes). It does not test them for fertility (ability to mate with other gametes). Viable gametes are not automatically functional (see previous and next comments).

Answer 2.5: We agree with this comment but we measure the fertility of the F1 hybrid, not of their progeny. Fertility typically refers to how many viable offsprings an individual can produce, irrespective of whether these individuals can themselves reproduce. All proxies of fitness are imperfect and do not measure all aspects of fitness. Spore viability is the most frequently used way of measuring strain fertility. We are using fertility in the context that others have in the literature, for instance in a recent paper by K. Wolfe (<https://journals.plos.org/plosbiology/article?id=10.1371/journal.pbio.2002128>) : ‘...interspecies hybrids can regain fertility, restoring their ability to perform meiosis and sporulate’ and by Greig et al., 2002 (<https://pubmed.ncbi.nlm.nih.gov/12061961/>) ‘Epistasis and hybrid sterility in Saccharomyces’. We agree that spores being viable does not mean that spores will be able to mate and produce viable progenies later on. However, one thing is sure, non-viable spores will not produce any descendants. Here we show that WGD leads to the first sterility barrier breakdown in hybrids, which is the first necessary step for fertility recovery. We agree that it would be interesting to examine this aspect further but this would be beyond the scope of this study which focuses on the rate and impact of WGD specifically. Furthermore, recent studies investigated the double sterility barrier breakdown in yeast synthetic hybrids as now developed in the discussion section (line 325-340).

“The double sterility breakdown in yeast hybrids could occur through two possible paths. The first one involves WGD by endoreduplication, which is a consequence of cytokinesis failure, followed by the loss of heterozygosity in the MAT locus in the corresponding diploid spores. The resulting spores would produce vegetative cells capable of mating type switching and mating to produce a fertile allopolyploid. This path to fertility recovery was recently observed in artificial Saccharomyces hybrids⁶⁸. The double sterility breakdown could also occur by means of damage to one copy of the mating type (MAT) locus in the diploid hybrid. This damage to the MAT locus could cause hybrid cells to behave as a haploid, switch mating type and autotetraploidize. This path to fertility recovery was recently described in natural hybrid species of the Zygosaccharomyces genus^{69,70}. However, in our experiment, mating type switching may not occur using the standard process because the necessary HO gene was deleted. The main way by which autotetraploidization could occur by mating in our study would be to have two hybrids with damage to the opposite MAT loci that are in the same colony and are close enough to mate with each other. Our data shows that

*this latter process is very unlikely since we did not detect genetic variations in chromosome III leading to such event and endoreduplication is the most likely mechanism of WGD in our experiment*⁴⁶.”

2.6 The observed instability of large chromosomes may be (parental) strain-specific because other works have reported on high instability of small chromosomes. The instability of Chr III is especially important because its malsegregation at meiosis can break down the MAT heterozygosity. Loss of MAT heterozygosity makes the gametes (ascospores) of the allotetraploid fertile. Again, the comparison of the authors' own observations (unstable large chromosomes) with the results published in previous works (unstable small chromosomes) could strengthen the paper.

Answer 2.6: In our study we found a higher frequency of small chromosome loss in diploid intra-lineage crosses and hybrids, as previous works did. However, we found a higher frequency of larger chromosome loss in tetraploids. Most studies on aneuploidy frequency by chromosome in yeast concerned diploid *S. cerevisiae* or few hybrids and generally gains and losses are not calculated separately. However massive chromosome loss is frequently observed in polyploids including large chromosome loss.

We now added a paragraph on this in the discussion (line 376-392):

*“Interestingly, we find different patterns of chromosome loss in tetraploids compared to diploids. Chromosome loss, consistent with what was observed in *S. cerevisiae* diploids^{18,56,66} (for review⁶⁷) and meiotic products of intra-specific *S. paradoxus* crosses⁷², more frequently affects the smallest chromosomes (I and III) in diploid intra-lineage crosses and hybrids, probably because it affects a smaller number of genes. However, losses of larger chromosomes are observed in tetraploids, suggesting a less deleterious effect of their loss due to the multiple copies present in the cell and a lowered effect of genome imbalance compared to a large chromosome loss in diploids. Furthermore, we find that one of the allopolyploids (H2 hybrid) shows progressive reduction in ploidy from 4n to about 2.8n by large chromosome losses. Interestingly, this takes place from the same parental subgenome (*S. paradoxus*). This strain also shows the most markedly increased growth rate among tetraploid lines, suggesting that these large chromosome losses may provide fitness gain to hybrids. Ploidy reversion by chromosome loss was previously observed in yeast auto-tetraploids⁷¹ and allopolyploids^{19,29,31,73}. The exact mechanism by which this reduction occurs in auto-polyploids is unknown¹. However, a recent study showed that the chromosome loss from one of the parental subgenomes could be the consequence of cytonuclear incompatibilities in an allopolyploid frog⁷⁴. Increased GIN in polyploids could lead to ploidy reversion by chromosome loss as a way for genome stabilisation^{19,31}.”*

2.7 Postzygotic instability of genomes is also a known phenomenon in yeast interspecies hybrids. In spite of this, the authors refer in line 196 to hardly relevant papers on cancer cells which are neither yeast cells nor allopolyploid hybrids.

Answer 2.7: We agree that genomic instability is a known phenomenon in yeast interspecies hybrids. We now added more references on GIN in yeast interspecies hybrids (Line 214)

(Morales and Dujon, 2012; Dion-Côté and Barbash, 2017; Marsit et al., 2017; Sipiczki 2018; Lopandic 2018; Morard et al., 2020). However genomic instability is also well known and studied since a very long time in cancer cells and these references concern specifically polyploidy as a hallmark of GIN which is well known in cancer cells.

2.8 Many previous papers reported on the genetic instability and segregation of allotetraploids. Chromosomes can be lost (usually and preferentially from one of the subgenomes) both at mitotic and at meiotic divisions resulting in broad spectra of segregants of chimeric (mosaic) genomes. The authors' results described in lines 141-150 are consistent with the previous observations which should be mentioned. It would be even more useful to compare the authors' findings with previous ones to show how their results contribute to the better understanding of the postzygotic evolution of the hybrid genomes.

Answer 2.8: We thank the reviewer for these supporting observations. We now added this in the discussion (line 383-392):

*“Furthermore, we find that one of the allopolyploids (H2 hybrid) shows progressive reduction in ploidy from $4n$ to about $2.8n$ by large chromosome losses. Interestingly, this takes place from the same parental subgenome (*S. paradoxus*). This strain also shows the most markedly increased growth rate among tetraploid lines, suggesting that these large chromosome losses may provide fitness gain to hybrids. Ploidy reversion by chromosome loss was previously observed in yeast auto-tetraploids⁷¹ and allopolyploids^{19,29,31,73}. The exact mechanism by which this reduction occurs in auto-polyploids is unknown¹. However, a recent study showed that the chromosome loss from one of the parental subgenomes could be the consequence of cytonuclear incompatibilities in an allopolyploid frog⁷⁴. Increased GIN in polyploids could lead to ploidy reversion by chromosome loss as a way for genome stabilisation^{19,31}.“*

2.9 Preferential pairing of chromosomes within the subgenomes was recently investigated in *Saccharomyces* interspecies hybrids. The phenomenon is called allosyndetic pairing (or autodiploidisation of the allotetraploid meiosis). The terms are borrowed from plant genetics, but in contrast to plants, it does not restore fertility but maintains the sterility by producing non-fertilising MATa/MATalpha allodiploid gametes. Thus, we know much more about it than described as a hypothesis in the (almost one-hundred-year old) reference cited in the lines 151-159.

Answer 2.9: Indeed previous works by us and others showed that WGD breakdown the first sterility barrier of hybrids due to the allosyndetic pairing. However, the goal of this part is to investigate the impact of WGD on fertility of intra-lineage crosses. Darlington first hypothesised that during meiosis in intra-lineage crosses the occurrence of multivalent pairing between the four identical homologous chromosomes will prevent the correct segregation of chromosomes and hence reduce the fertility of intra-lineage crosses. In this part we aimed to verify if this is the case in our experimental model. Contrary to the hypothesis there is no difference between the spore viability of diploids and tetraploids in the VL_C intra-lineage crosses. However, half of the generated spores will be indeed heterozygous in the mating type locus which will reduce their ability to mate.

We now added a discussion about this (line 342-351):

“Contrary to the prediction of Darlington ⁴⁸, WGD has no negative effect on spore viability of intra-lineage crosses, which remains very high and similar to diploid fertility after WGD. Our results, in agreement with what was observed in some flowering plants ⁵⁴, suggest that although four identical copies of chromosomes are present in individual yeast cells, pairing and segregation seems to occur correctly during meiosis in auto-polyploid yeasts. However, WGD would create a second sterility barrier in half of the corresponding spores which will have heterozygous mating type loci. These diploid spores will however be able to sporulate and generate haploid spores competent for mating. Furthermore, tetraploids show a larger number of aneuploidies compared to diploids at the end of our experiment, which in the long term could decrease the spore viability of auto-tetraploids.”

2.10 The terms di-, tetra- and aneuploid should be used consistently and clearly. E.g. if a tetraploid hybrid loses a chromosome or a pair of homologous chromosomes, it is no longer tetraploid. It is a trisomic or a disomic aneuploid segregant/derivative of the allotetraploid hybrid.

Answer **2.10**: We now modified this (line 233-234):

“These crosses are trisomic or tetrasomic and have one or two additional copies of chromosome XII”

2.11 Please be more specific about LOH. Loss of heterozygosity can be due to the loss of an entire chromosome (or a pair of homologous chromosomes in allotetraploids) from one subgenome or to (the much less likely) gene conversion/recombination between allosyndetic (homeologous) chromosomes of the subgenomes. Is there any correlation between the loss of chromosomes and the loss of heterozygosity at genes located on those chromosomes?

Answer **2.11**: Here the rate of Loss of heterozygosity that we measured concerns the rate of gene conversion and recombination between allosyndetic chromosomes. The rate of chromosome loss leading either to a whole chromosome loss of heterozygosity or to a change in allele frequency in polyploids is calculated separately since distinct molecular processes are behind these two mechanisms of genetic variability. We separated the LOH events resulting from mitotic recombination to interstitial and terminal LOH. We added more details about that in the text.

(line 223-225) “We measured the rates of aneuploidy (chromosome gain and loss) and loss of heterozygosity (LOH) resulting from mitotic recombination within chromosomes, two typical GIN hallmarks ^{57,58}.”

(line 286-287) “Tetraploids also show non-significant differences with diploids in LOH (most likely resulting from mitotic recombination) rate”

2.12 Timing of genome duplication. Lines 137-138: „Genome doubling occurred at different generation time points after mating; some of them emerged quickly, in less than 90

generations, while others appeared after more than 680 generations“. Wouldn't be it more accurate to say that the number of tetraploid cells reached the threshold of detectability after XX generations? Can you exclude the possibility that the genome-doubling event or events took place much earlier than the e.g. 90th generation and became detectable much later, only after they had grown to a detectable subpopulation (by e.g. the 90th generation) (provided they were competitive!!!).

Answer 2.12:

The reviewer is correct that WGD events could have occurred before but not fixed in a colony or occur in a colony that was not used to create the next round of colonies. However, we cannot know if it happened or not. This is the nature of the MA experiments, which are not perfect but so far have been the best way of measuring mutation rates. The protocol is established so that all mutations have the almost same probability of fixation, unless they are extremely deleterious. In this case, fixation refers to line loss. We can therefore only talk about events that were fixed during the experiments. Because all mutations have a similar probability of fixation, we can conclude that the rate at which we observe them is proportional to their rate of occurrence. We understand the concern of the reviewer and we should not have used 'quickly' in this sentence. We now use early rather than quickly.

We now write (line 151-152):

“Genome doubling occurred at different time points after mating; some of them emerged early, before the 90th generation, while others appeared after more than 680 generations.“

2.13 Line 114 „within 770 generations, 0 to 11% of the lines went through spontaneous WGD.“ This formulation is somewhat confusing. Do the authors want to say that 11 % of the diploid hybrid cells converted to tetraploidy (millions of independent genome doubling events) by the time of examination or they mean that 11% of the population was tetraploid at the time of examination. I doubt that the former can be measured. If the latter was meant by the authors then they cannot say that the percentages of tetraploids reflect the frequency of (disposition to) genome duplication because even a high percentage can be the result of a single duplication event if the allotetraploid is more competitive than the allodiploid in the applied culturing conditions. Its percentage may simply reflect the time point of the event: the higher the percentage, the earlier the event.

Answer 2.13: WGD rates are listed in the Supplementary Table 2 and we now changed the sentence (line 127-128):

“This shows that within 770 generations, spontaneous WGD occurred in 0 to 11% of the populations among the different crosses”

2.14 What is the difference between the pseudohaploid and the diploid cell (line 124)? If the genome is diploid (due to e.g. endomitosis), the cell is diploid regardless of the homo-/heterozygosity at the MAT locus. To call it pseudohaploid may cause confusion. Mating competence is not a privilege of haploids.

Answer **2.14**: We now changed pseudohaploids to diploids competent for mating (line 138-140):

“They most likely arose from mating between haploid SpB strains and diploidized and competent for mating SpC strains that arose from haploid stocks⁴⁶.”

Etc.

2.15 Lists of publications relevant for the comments can be found in recent reviews: e.g. doi.org/10.1002/yea.3294, 10.3389/fmicb.2018.03071, 10.1093/femsyr/foaa040

Answer **2.15**: We thank the reviewer for providing these references that now are mentioned in the text.

REVIEWERS' COMMENTS

Reviewer #1 (Remarks to the Author):

I think the revised version of the manuscript is much improved and more fully able to support the study claims. In particular, the expanded discussion of previous research in interspecific yeast hybrids helps to place the study in context. I think the study is a valuable contribution to our understanding of complex mechanisms of hybrid evolution and speciation. I have no further comments to improve the manuscript at this time.

Reviewer #2 (Remarks to the Author):

The authors have addressed most of the concerns I had with the previous submission. I only have a minor comment to Response 2.2. ("...we do not know of any studies that have measured the rate of WGD to demonstrate experimentally that hybridization can trigger WGD...") When larger numbers of hybrids were analysed, genome duplication was detected in the vast majority of synthetic *S. uvarum* x *S. cerevisiae* and *S. uvarum* x *S. kudriavzevii* hybrids. In fact, stable synthetic allodiploids were the rare cases. Relevant publications are listed in Ref. 31 (e.g. doi: 10.1007/s00253-017-8274-9). This fact must have been overlooked by the authors and this is why I said that the title is somewhat misleading. A reference to these previous observations would not reduce the value of the work presented. On the contrary, it could be another argument justifying that this phenomenon is worth examining.

REVIEWERS' COMMENTS

Reviewer #1 (Remarks to the Author):

I think the revised version of the manuscript is much improved and more fully able to support the study claims. In particular, the expanded discussion of previous research in interspecific yeast hybrids helps to place the study in context. I think the study is a valuable contribution to our understanding of complex mechanisms of hybrid evolution and speciation. I have no further comments to improve the manuscript at this time.

Answer: We thank the reviewer for the very favorable review.

Reviewer #2 (Remarks to the Author):

The authors have addressed most of the concerns I had with the previous submission.

Answer: We thank the reviewer for the favorable review.

I only have a minor comment to Response 2.2. (“...we do not know of any studies that have measured the rate of WGD to demonstrate experimentally that hybridization can trigger WGD...”.) When larger numbers of hybrids were analysed, genome duplication was detected in the vast majority of synthetic *S. uvarum* x *S. cerevisiae* and *S. uvarum* x *S. kudriavzevii* hybrids. In fact, stable synthetic allodiploids were the rare cases. Relevant publications are listed in Ref. 31 (e.g. doi: 10.1007/s00253-017-8274-9). This fact must have been overlooked by the authors and this is why I said that the title is somewhat misleading. A reference to these previous observations would not reduce the value of the work presented. On the contrary, it could be another argument justifying that this phenomenon is worth examining.

Answer: We now added these references (lines 68-70):

“Polyploidy is also commonly observed in recent human associated (Querol and Bond 2009; Morales and Dujon 2012; Marsit et al. 2017; Lopandic 2018; Gabaldón 2020) and synthetic (Pfliegler et al. 2012; Karanyicz et al. 2017; Sipiczki 2018) yeast hybrids, suggesting that allopolyploidization might not be a rare mechanism of yeast hybridization (Gabaldón 2020).”

We now changed the title to:

“The neutral rate of whole-genome duplication varies among yeast species and their hybrids”